# Weight Regain after Metabolic Surgery: Beyond the Surgical Failure

**DOI:** 10.3390/jcm13041143

**Published:** 2024-02-18

**Authors:** Juan Salazar, Pablo Duran, Bermary Garrido, Heliana Parra, Marlon Hernández, Clímaco Cano, Roberto Añez, Henry García-Pacheco, Gabriel Cubillos, Neidalis Vasquez, Maricarmen Chacin, Valmore Bermúdez

**Affiliations:** 1Endocrine and Metabolic Diseases Research Center, School of Medicine, University of Zulia, Maracaibo 4004, Venezuela; 2Departamento de Endocrinología y Nutrición, Hospital Quirónsalud, 28009 Madrid, Spain; 3Facultad de Medicina, Departamento de Cirugía, Universidad del Zulia, Hospital General del Sur, Dr. Pedro Iturbe, Maracaibo 4004, Venezuela; 4Unidad de Cirugía para Obesidad y Metabolismo (UCOM), Maracaibo 4004, Venezuela; 5Clinica Obesidad y Envejecimiento SAS, Bogotá 110111, Colombia; 6Facultad de Ciencias de la Salud, Universidad Simón Bolívar, Barranquilla 080001, Colombia; 7Centro de Investigaciones en Ciencias de la Vida, Universidad Simón Bolívar, Barranquilla 080001, Colombia

**Keywords:** metabolic surgery, obesity, complications, metabolic syndrome, weight

## Abstract

Patients undergoing metabolic surgery have factors ranging from anatomo-surgical, endocrine metabolic, eating patterns and physical activity, mental health and psychological factors. Some of the latter can explain the possible pathophysiological neuroendocrine, metabolic, and adaptive mechanisms that cause the high prevalence of weight regain in postbariatric patients. Even metabolic surgery has proven to be effective in reducing excess weight in patients with obesity; some of them regain weight after this intervention. In this vein, several studies have been conducted to search factors and mechanisms involved in weight regain, to stablish strategies to manage this complication by combining metabolic surgery with either lifestyle changes, behavioral therapies, pharmacotherapy, endoscopic interventions, or finally, surgical revision. The aim of this revision is to describe certain aspects and mechanisms behind weight regain after metabolic surgery, along with preventive and therapeutic strategies for this complication.

## 1. Introduction

The World Health Organization (WHO) defines overweight and obesity as an abnormal, excessive and harmful fat accumulation because it is a well-known independent risk factor for morbid conditions like diabetes mellitus (DM), dyslipidaemia, cardiovascular diseases, and cancer [1]. Over the last four decades, the prevalence of obesity has increased at an alarming rate in countries with Westernized lifestyles, becoming one of the major health concerns as a consequence of morbidity, mortality and the economic burden on national healthcare systems worldwide [2,3,4,5]. In fact, since 1975, obesity prevalence has shown a three-fold increase in adults and a five-fold increase in children and adolescents. Furthermore, according to the latest regional and national projections by the 2023 World Obesity Atlas report on obesity, the majority of the global population (51%, or over 4 billion individuals) will be suffering from being overweight or obesity, defined as a Body Mass Index (BMI) ≥ 25 kg/m^2^ and BMI ≥ 30 kg/m^2^. If current trends continue, the global economic impact of excess weight could reach $4.32 trillion annually, equivalent to 3% of the global GDP [6,7].

Despite multiple therapeutic strategies for weight loss (WL), combining several nutritional schemes, physical activity, cognitive-behavioral therapy, and pharmacologic intervention, medical obesity management is a challenging endeavor, often yielding limited success, as lifestyle-based interventions alone prove insufficient in achieving significant long-term weight loss in some patients, occasionally leading to a rebound effect where individuals regain more weight than initially present at the beginning [8,9].

On the other hand, the drugs approved for obesity are orlistat, phentermine/topiramate, naltrexone/bupropion, and the Glucagon-like peptide receptor agonists, like Liraglutide or Semaglutide, although these treatments can be expensive and may have adverse effects. Thus, it is important to carefully consider the potential benefits and risks of drug therapy before starting treatment in individuals with obesity [10]. It is important to mention that extensive pharmacological advances have been made, which will be discussed in greater depth below.

Given this challenge, along with the need for an effective and long-lasting treatment, metabolic surgery (MS) has proved its efficacy in losing massive amounts of both subcutaneous and visceral fat [11], which involves different techniques designed to correct or control obesity, aiming to improve quality of life by achieving adequate and long-lasting WL with minimal complications [12].

In 2022, the American Society for Metabolic and Bariatric Surgery (ASMBS) and International Federation for the Surgery of Obesity and Metabolic Disorders (IFSO) states that MS is recommended in case of: BMI ≥ 35 kg/m^2^ (regardless of presence, absence, or severity of co-morbidities), patients with T2D and BMI ≥ 30 kg/m^2^, individuals with BMI of 30–34.9 kg/m^2^ who do not achieve weight loss or co-morbidity improvement using nonsurgical methods. Also, it is important to consider geographic factors; for example, obesity in Asiatic people is recognized as BMI > 25–27.5 kg/m^2^, so MS could be performed in these cases. On the other hand, age is not an exclusion or inclusion criteria for MS, and could be performed in Children and adolescents with BMI > 120% of the 95th percentile and a major co-morbidity, or a BMI > 140% of the 95th percentile [13,14,15,16].

MS procedures had traditionally been divided into restrictive, malabsorptive and mixed categories; however, it is now known that MS can cause weight loss not only through these mechanisms but also through appetite control, alterations of hormones of the gut brain axis, alterations in bile acid physiology and intestinal microbiota [17,18,19,20]. Even though MS successfully manages to decrease a significant percentage of body weight, not all patients can maintain the weight loss achieved and could surprisingly regain the lost weight [21]; this undesired scenario could affect the patient’s physical and mental health. Recently, this weight regain (WR) has been contemplated as a complication of multifactorial etiology; therefore, the current review will describe the main features and mechanisms behind WR together with preventive and therapeutic strategies [12,22].

## 2. Metabolic Surgery Complications

As previously mentioned, MS is a low-complication rate procedure with a minimum margin of risk. Overall, the combination of improved surgical techniques, surgeon expertise, patient selection, perioperative care, post-operative management, and advancements in technology and research have resulted in a notable decrease in the incidence of complications linked to bariatric surgery. Nonetheless, it is not exempt from complications, and MS has been associated with adverse surgical complications, including high-mortality complications [22]. Furthermore, Pallati et al. [23], in a systematic review and meta-analysis of 160,000 bariatric patients, reported a post-operative complications rate between 10–17% and a 7% reoperation rate; favorably, the mortality rate remained low (0.08–0.35%).

The perioperative or short-term complications can be divided into minor and major; the most common minor complications are usually at the surgical site (port bleeding or hematoma, skin infections, and post-operative neuropathic pain), hydro electrolyte imbalance, and urinary tract infections. Major complications involve anastomotic leaks, intra-abdominal bleeding, small bowel perforation, myocardial infarction, and pulmonary embolism <30 days after MS. Typically, these early complications frequency is below 1.6%, and the mortality rate is <0.7% [22]. Post-surgical leaks can arise from the gastrojejunal anastomosis of the RYGB (1.68–2.05%); in vertical sleeve gastrectomy, they emerge from the staple line (2.2%). Hemorrhages often begin in the staple line but can also come from anastomotic or gastric remnant ulcers [24].

Moreover, although mid- and long-term complications have been well described, establishing their exact incidence is difficult due to the increasingly significant number of patients who miss their follow-up visits as time goes by. These complications are stenosis, bowel obstruction, marginal ulcers, ventral hernia, fistula, gastroesophageal reflux disease, and metabolic complications like nephrolithiasis and hypoglycemia [25]. In this context, gastrojejunal stenosis is a common complication, with an incidence rate ranging from 4% to 27%, similar to gastroesophageal reflux disease, which occurs in 12% of cases. Meanwhile, gastric stenosis is uncommon, only occurring in 1% of the patients. Internal hernias generally cause small bowel obstructions after gastric bypass or rarely by intraperitoneal adhesions in 2–3% of the patients [25].

Several kinds of ulcers, most marginal, arise within the first 12 months after gastric bypass; their estimated incidence is around 16%, contrasting the much lower incidence of fistulas (1.2%). Unfortunately, fistulas can appear in almost any location of the digestive tract following surgery; gastro-gastric fistulas are an especially alarming RYGB complication. Another rare complication is hernias, which have a frequency of less than 1% in laparoscopic procedures; however, the frequency increases to around 8% for open procedures [24].

Additionally, vitamin and mineral deficiencies are also described as possible complications. Regarding nutritional deficiencies, biliopancreatic diversion leads to a more significant decrease in liposoluble vitamins, copper, and zinc compared to gastric bypass; contrarily, vitamin B12 deficiency, due to decreased levels of its intrinsic factors, is more frequently caused by gastric bypass in comparison to any other procedure [26].

In contrast to these low surgical complications, 10–20% of the patients are expected to regain a significant weight proportion in the long term. Likewise, it has been reported that 20–25% of the lost weight can be regained in a ten-year course, starting nearly 24 months after the surgery; thus, the mean weight regained after the surgery is 10 kg, ranging from 0.5 to 60 kg [27]. In this context, WR is considered clinically significant when WR is greater than or equal to 15% of the lowest weight reached while maintaining this increase for at least six months [28].

## 3. Risk Factors for Weight Regain after MS: Is It All about the Surgery?

The multifactorial nature behind WR after MS has hampered the comprehension of its mechanisms, and thus, the therapeutic approach. Psychological, behavioral, endocrine metabolic, genetic, and anatomical factors have been associated with WR [29,30].

### 3.1. Anatomic and Surgical Factors

The leading anatomical abnormality associated with WR is the enlargement of the gastric pouch (>6 cm long or >5 cm wide) and gastrojejunal stoma (diameter > 2 cm), and gastro-gastric fistula (GGF), which are mainly sequelae of procedures such as RYGB and vertical sleeve gastrectomy (VSG) [31,32,33]. A GGF is an abnormal communication between the proximal gastric pouch and the distal gastric remnant. Consequently, food detours to the “previous route” instead of the duodenum, increasing the available gastric volume and food’s absorption surface impairing the properties of MS [34,35,36].

By contrast, gastrojejunal stoma dilation leads to accelerated gastric pouch emptying and, therefore, a lack of satiety, instead accommodating larger amounts of food within the gastric pouch. Heneghan et al. [32] assessed the potential causes of WR by gastroscopy in patients submitted to RYGB (*n* = 380), reporting that only 28.8% of those who had WR (*n* = 205) had a normal-sized stoma, contrasting to 63.4% in patients who had successful weight loss (*n* = 175). Simultaneously, univariate statistical analysis demonstrates that the length and dilation of the stoma are the most influencing factors of WR; interestingly, the multivariate analysis only found the latter to be an independent factor for WR. Similarly, a retrospective study carried out by Yimcharoen et al. [33] reported that out of 205 patients with WR after RYGB, 58.9% had dilation of the gastrojejunal stoma, 28% had enlargement of the gastric pouch, and 12.3% had both of these findings.

Similarly, a study in people submitted to RYGB reported that limb length does not influence post-MS weight changes [37]. Still, this could be attributed to the methodology, study sample size, and confounding variables that could induce WR and enlargement of the gastric pouch, such as patients’ lifestyles and psychological status.

### 3.2. Endocrine and Metabolic Factors

RYGB has been associated with episodes of hypoglycemia, a significant clinical component of the dumping syndrome (DS). In a study involving 36 RYGB patients, Roslin et al. [38] assessed their glucose levels six months after the procedure. They found that 11 patients had weight regain exceeding 10%, and six of these experienced hypoglycemia two hours after glucose load. Likewise, a study performed by Varma et al. [39] on 428 American patients who underwent MS determined that the odds of WR were significantly higher in those who had symptoms of hypoglycemia (OR: 1.66; 95% CI: 1.04–2.65). The authors have suggested that the causal relation could be due to metabolic changes produced by glucose homeostasis effects on appetite and gastrointestinal functioning [39]. Additionally, it has been proved that in the long term, post-MS patients with WR exhibit alterations in the levels of gastrointestinal and neuronal peptides related to appetite and satiety, which could indicate that changes in hormonal parameters contribute to WR [40].

### 3.3. Lifestyle: Eating Patterns and Physical Activity

Implementing lifestyle changes that counter the “obesogenic” environment before surgery is essential to achieve meaningful results in the WL after MS. In this context, it has been described how post-MS patients regain weight by eventually neglecting their lifestyle changes [41]. Furthermore, once the patients reach their target weight, some may increase their caloric intake, an expected eating behavior two years after the surgery [42,43].

Some of patients either fail to keep adequate control of their post-MS nutritional status or refuse entirely to follow the dietary patterns suggested by the weight management team, contrarily, maintain a high caloric intake attributed to a large intake of high-fat food, junk food, sweets, and high-sugar drinks, leading to suboptimal WL or even WR [41,43,44]. Bassan et al. [44,45] conducted a retrospective study that included 80 patients at least 24 months after MS and reported that 23.7% presented a WR greater than 10% of the lowest post-operative weight. Moreover, supported by multivariate analysis, a positive association between Healthy Eating Index and WR was observed (OR 0.95; *p* = 0.04), correlating to similar results found by other authors [44,46].

Furthermore, common maladaptive eating patterns among post-MS patients, such as binge eating and grazing, are considered risk factors for reduced WL [47]. Grazing can be defined as repeated episodes of consuming small quantities of food over a long time and is usually accompanied by feelings of guilt and loss of control [47,48]. A systematic review including 994 post-MS patients showed 16.6–46.6% engaged in grazing and 47% engaged in WR; an association was found between these two variables regardless of the type of MS and the author’s definition of grazing [49].

Another factor linked to WR is dysphagia, a frequent complication after RYGB [50]. In a prospective cohort study by Runge et al. [51] on 245 post-MS patients, a higher WR was observed in those with dysphagia (*n* = 49) in comparison to the control group (*n* = 196) (37% vs. 25%). These things considered, patients with dysphagia are more likely to incline their diet partially or even entirely towards soft or liquid food since they are absorbed faster and produce less satiety, favoring a higher caloric intake and thus a positive energy balance that could explain the WR [51,52].

Regarding the food preferences observed after MS, although patients reported a diminished explicit liking for sweet foods at 3 months post-surgery and a lower desire to consume them at both 3- and 12-months post-surgery, intake of high-sugar foods was maintained in another study [53]. In this regard, a meta-analysis showed that bariatric surgery could be effective on energy and fat intake; however, there was no effect on carbohydrate intake [54], being considered another risk factor for developing weight gain after MS.

Besides unhealthy dietary habits, sedentarism or lack of physical activity can also be risk factors for WR. According to Rosenberg et al. [55], only 10–24% of the patients who underwent MS had performed the minimum physical activity to maintain their health status. Moreover, in a study by Yanos et al. [56] on 97 patients submitted to RYGB, 26% exhibited WR, associated with nocturnal eating, alcohol consumption, and diet and physical activity modifications. Correspondingly, Freire et al. [44] reported a lower incidence of WR among post-MS patients who exercised than those who did not.

### 3.4. Psychological Factors and Mental Health

Neuropsychiatric and Psychological disorders have been linked to WR and can hinder adherence to dietary and behavioral intervention plans during and after MS [46]. Binge eating disorder (BED), defined by The Diagnostic and Statistical Manual of Mental Disorders-Fifth Edition (DSM-5) as the uncontrolled consumption of larger and more than usual food quantities within two hours, is one of the main predictors of WR after MS [57]. However, BED prevalence among post-MS patients varies significantly according to the criteria used by different authors, ranging from under 5% to almost 24% [58,59]. Still, it has been demonstrated that BED is related to reduced WL or even WR two years after the procedure, along with developing worse maladaptive eating behaviors than prior MS [47,60]. Similarly, other eating disorders, such as soft food and night eating, have been recognized as predictors of WR after MS [61].

Furthermore, psychiatric disorders increase the risk of WR during post-operative periods; for instance, Rutledge et al. [62] showed that those individuals presenting two or more psychiatric disorders were six times more likely to develop WR post-MS. Under this framework, depression stands out among the most common disorders in bariatric patients, and although the association between this and WR or failure in post-MS WL has been demonstrated, as well as its presence predisposes individuals to be more prone to develop eating disorders, the results of studies tend to contradict each other since some show that depression is diminished after MS or, failing that, no causal relationships are observed in their analyses [56,63,64,65,66]. Similarly, patients with WR have high clinical or borderline anxiety and stress levels; however, these were not associated with higher energy consumption [66,67].

Finally, drug use and alcoholism have been described as influential factors in WR, as post-MS patients may seek relief from other substances through “addiction transfer” to substitute the needs established by the brain reward system for excessive energy consumption prior to MS [60,68,69]. Odom et al. [65] followed up on 203 post-RYGB patients, showing that decreased post-MS well-being, increased need to eat, and preoccupation with drug or alcohol use (addictive behavior) were independent predictors of WR. Thus, it is clear that bariatric patients need pre- and post-MS psychological assessment to ensure expected outcomes in WL and avoid relapse in maladaptive habits related to WR [70].

### 3.5. Preoperative and Other Factors

Numerous studies have found preoperative factors that may predispose patients to WR. A prospective study of 782 bariatric patients showed that approximately 50% of them presented WR, with patients in the super-obese group having a higher percentage of surgical failure (18.8%) and WR. The authors concluded that individuals with higher BMI before surgery are more likely to WR [71]. In particular, Keith et al. [72], in their retrospective study, described that preoperative factors such as male sex (*p* = 0.020), white race (*p* < 0.001), and high socioeconomic level (*p* = 0.035) were associated with WR. Furthermore, when multivariate analysis was performed, it was observed that socioeconomically advantaged patients were more likely to have WR than the rest (OR: 1.82, CI 1.18 to 2.79). However, other authors have differed with this study since their analyses establish that female sex and black race could be considered risk factors for WR [37,73,74].

Although age has been highlighted as a possible preoperative factor related to WR, results among studies vary significantly, with both young and older adults (>60 years) reported to be prone to WR [73,75,76]. Moreover, time elapsed after surgery, iron deficiency [77], work activity related to eating, and comorbidities such as T2DM have been linked to WR [78].

Remarkably, medications for psychiatric disorders, including tricyclic antidepressants, valproic acid, lithium, and antipsychotics, as well as antidiabetic drugs, steroids, and contraceptives, have been associated with weight gain and positive appetite modulation [79]. In this regard, post-MS patients treated with any of these drugs could theoretically be at high risk for WR [80]. In addition, there are genetic factors related to the development of obesity and adipose tissue (AT) biology that could be implicated in post-MS WR. Among these, genetic polymorphisms of AT adrenoreceptors such as the *ADRB2* gene (*Gly16Arg* and *Gln27Glu*), and those related to leptin, such as *LEPR*-gene (*LEPR Lys109Arg*, *LEPR Gln223Arg*, *LEPR Lys656Asn*), stand out [81].

## 4. Underlying Mechanism of Weight Regain after Metabolic Surgery: Neuroendocrine and Metabolic Perspectives

The mechanisms involved in post-MS WR remain poorly understood and are mainly based on possible hormonal changes and the body’s response to caloric restriction following MS. Hypothetical models on neuroendocrine and metabolic mechanisms influencing WR in post-MS patients are presented below (Figure 1). However, the theoretical models will be grounded on the results of preclinical and clinical studies whose results tend to vary and be controversial, possibly due to all methodological aspects involved.

In this vein, gastrointestinal hormones and their interactions with the nervous system play a fundamental role in WR. The hypothalamus is the main energy regulator of physiology, appetite, and body weight. It maintains reciprocal communication with other central areas and peripheral organs through afferent and efferent fibers, neurotransmitters, hormones, and peptides [82,83]. In particular, the arcuate nucleus (ARC) has two neuronal populations with contrasting effects on appetite. In this respect, an orexigenic population promotes appetite by releasing neuropeptide Y (NPY) and agouti-related peptide (AgRP). On the other hand, anorexigenic neurons promote satiety by releasing pro-opiomelanocortin (POMC) metabolites such as melanocyte-stimulating hormone (α-MSH), together with cocaine and amphetamine-regulated transcript (CART) [84,85].

Other hypothalamic regions such as the paraventricular nucleus (PVN) and the lateral hypothalamic area (LHA) (where orexigenic peptides such as orexins 1 and 2 are released), as well as the nucleus of the solitary tract (NTS) and the postrema area (PA) of the brainstem, are part of the energy intake (EI) regulatory intercommunication network [86,87,88]. In addition, these areas are interconnected with the limbic system and regions of the cortex involved in reward control and stimulus perception, among other key functions for EI [89,90].

On the other hand, the gastrointestinal tract releases peptides according to nutrient availability; that is, under fasting or post-prandial conditions. In states of starvation or fasting, ghrelin is released by the X/A cells of the gastric fundus; this is a polypeptide of 28 amino acids that has its receptor (GHS-R) in several anatomic structures, including the vagus nerve and the hypothalamus. Ghrelin promotes appetite by activating NPY/AgRP and LHA orexigenic neurons [91,92,93,94] and possesses gastrointestinal prokinetic properties that promote gastric emptying. The latter mechanism enhanced appetite and EI by decreasing gastric distension and satiety signals [95,96].

Moreover, the mechanical and chemical stimuli produced by food intake, the formation of the food bolus, and its caloric content can lead to the release of anorexigenic gastrointestinal peptides involved in satiety, like peptide YY (PYY), cholecystokinin (CCK), glucagon-like peptide 1 (GLP-1), among others [83,97]. These peptides can counteract ghrelin activity by blocking its release, acting on anorexigenic POMC/CART neurons, and inhibiting NPY/AgRT signals in the hypothalamus. Therefore, anorexigenic hormones act either through direct action on receptors located in the hypothalamus (PYY binds directly to Y2 receptors) or by promoting peripheral mechanical satiety signals through vagal afferents that interact with areas such as the AP and NST (in the case of CCK) [98,99,100,101]. Moreover, these inhibitory signals by gastrointestinal peptides are reinforced by the activation of serotonergic 5-HT1b receptors in subcortical areas, which promotes the suppression of orexigenic pathways [102].

After RYGB, hormonal changes that promote WL are observed, mainly an increase in PYY levels and a decrease in ghrelin levels compared to obesity or pre-surgery subjects [103,104,105,106,107]. However, preclinical and clinical studies show that these hormone levels are affected in mice and individuals with post-RYGB and, curiously, post-VSG WR. In fact, ghrelin and PYY levels are similar to those prior to surgery or not high enough compared to individuals with long-term WL. Thus, there is a higher proportion of orexigenic peptides than anorexigenic peptides [40,108,109,110]. Accordingly, these changes could increase appetite and IE, correlating with maladaptive eating patterns in postbariatric patients [46]. This theory becomes relevant in patients with GGF and other post-MS anatomical alterations. GGF can increase ghrelin secretion by X/A cells in the sectioned stomach in these cases. Curiously, the levels of orexigenic hormones and the subjects’ weight are diminished after GGF correction and gastric stoma lengthening [111,112].

Another peptide that affects the metabolic status of post-RYGB patients is GLP-1. It has insulinotropic properties that contribute to the remission of metabolic comorbidities such as T2DM [113,114]. Studies have found that GLP-1 levels increase after RYGB [107,115], leading to a hyperinsulinemic response that may result in postprandial reactive hypoglycemia (a characteristic of DS), along with accelerated gastric emptying that may decrease satiety-related mechanical distention signals [38,116]. Since glucose is recognized as a modulator of appetite, low levels of this molecule following a hyperinsulinemic response may promote an increase in EI and WR [38,39].

Furthermore, post-MS caloric restriction is related to changes in the metabolic profile of AT, favoring adipogenesis and decreasing both adipocyte size and insulin sensitivity. These changes could augment glucose uptake by AT, which would generate increased lipogenesis and fat storage, contributing to adipocyte hypertrophy, a hallmark in patients with obesity [117,118,119,120]. In this sense, bariatric patients could maintain a dysfunctional AT, a cornerstone of deflective endocrine metabolic signals involved in WR-related hyperphagic states [118]. These mechanisms would support the hypothesis of the obesogenic memory of immune cells [121].

In this sense, dysfunctional AT is characterized by a proinflammatory secretory profile. One of the principal adipokines secreted by the dysfunctional AT is leptin, a hormone related to satiety control [122]. Leptin levels are proportional to the degree of energy storage; i.e., in patients with high levels of adiposity, leptin release is higher [123,124]. This increase leads to leptin resistance in the hypothalamus due to receptor downregulation [125,126]. Contrarily, lower leptin levels have been observed in mouse models under calorie-restricted diets and following RYGB compared to controls or cases where surgery was defective [127,128]. These results may reflect a compensatory mechanism to the energy deficit that promotes hunger for energy conservation [88,129,130].

Likewise, low leptin levels are associated with the production of endogenous endocannabinoids. These compounds bind to CB1 receptors in the hypothalamus and AT, triggering the secretion of orexigenic factors and lipogenic activity, respectively [131,132,133]. Similarly, other homeostatic mechanisms that seek to reestablish a state of energy balance have been described, including a reduction in thyroid and sympathetic activity due to low leptin levels. These homeostatic responses could promote lower resting energy expenditure, increasing appetite and energy replenishment [128,130,134,135]. In addition, these changes in energy and leptin levels can stimulate the PVN, favoring hypothalamic-pituitary-adrenal axis activity and thereby increasing cortisol release. High plasma cortisol concentrations are related to increased appetite and fat storage [136,137,138]. However, the effects of leptin on WR remain controversial [40,139,140].

Energy balance is essential for maintaining WL or, conversely, WR. Following MS, there is a depletion of EI that will eventually decrease basal metabolism and resting energy expenditure. These metabolic changes generate a negative energy imbalance. Consequently, our body seeks to respond to this deficit by promoting EI and WR [118,141,142,143,144]. This “energy gap” can be counteracted by physical activity, which is why it is always recommended to implement a physical exercise regimen along with the diet to achieve WL and its long-term maintenance [144,145].

In contrast, non-homeostatic WR-related mechanisms could be extrapolated to post-MS patients [146,147]. Brain areas of the mesocorticolimbic system play a key role in the reward system, eating behavior, and perception of sensory stimuli that may affect EI [89,90,148]. Studies using positron emission tomography and magnetic resonance imaging have described the activation of some of these regions in individuals who, after WL, responded positively to food stimuli that promoted motivation and desire to eat energy-rich foods, rectifying the role of hedonic pathways in WR [146,149,150]. At the same time, hormones such as ghrelin, leptin, and endogenous endocannabinoids constantly interact with the positive reinforcement of the reward system, so post-MS individuals likely relapse into previous high-calorie consumption patterns [151,152,153,154].

It should be emphasized that further research should be carried out despite the possible neuroendocrine, metabolic, and adaptive theories previously presented. Preclinical and clinical studies should be promoted to elucidate the exact mechanisms involved in WR and to establish an appropriate approach to combat the development of this complication of MS.

## 5. Strategies for Prevention and Management of Weight Regain after Metabolic Surgery

WR post-MS is a challenging and complex phenomenon that urges specialists to design an ideal plan to prevent and treat this condition. Due to the multifactorial component of WR, several studies have been conducted that consider various therapeutic perspectives to ameliorate WR. One is a specialized multidisciplinary care approach involving behavioral therapies, lifestyle changes, and weight management [155,156,157,158,159,160]. Other approaches under study include the use of appetite suppressants or anorexigenic pharmacotherapy [156,161] (Table 1), as well as endoscopic procedures and even revision surgery (Table 2) [162,163,164,165], considered as one of the last options due to its well-known complications [166].

### 5.1. Lifestyle Changes

First, according to several studies, assessing MS patients through a weight control program is a good approach to preventing WR [156]. In this regard, recognizing mild (0.2% to <0.5%), moderate (0.5% to 1.0%), and rapid (>1.0%) WR within 30 days is essential for early detection of this condition [60], finding that routine long-term clinical follow-up of the patient is necessary.

Recently, researchers have evidenced that in addition to having a weight control program, the establishment of healthy dietary and physical activity (PA) behaviors are good predictors of WL post-MS [167]. In a 2-year follow-up study, Nuijten et al. [157] found better body composition and quality of life in bariatric patients who achieved changes in their PA. Likewise, a prospective observational study linked higher WL to individuals who adhered to Mediterranean dietary patterns [158]. Because of the relationship between healthy behaviors and better post-operative outcomes, a multidisciplinary care program with weight control and lifestyle interventions is necessary to prevent and treat WR post-MS [159].

In another context, post-operative behavioral management was postulated as an emerging strategy for post-RYGB WR in a meta-analysis [168]. In this sense, a pilot study demonstrated the feasibility of the behavioral intervention based on acceptance and WL, since after ten weeks, WR managed to stop and reverse with an overall mean of 3.58 ± 3.02% of total body weight (TBW) loss among individuals who completed the treatment [180]. Similar findings were evidenced in post-BGYR WR patients, in which weight (mean 1.6 ± 2.38 kg), depressive symptoms (*p* ≤ 0.01), grazing patterns (*p* ≤ 0.01), and binge eating episodes (*p* ≤ 0.03) decreased significantly after behavioral intervention lasting six weeks [160].

Likewise, the support of a remotely applied behavioral therapy to help patients with WR was carried out by the Healthy Eating and Lifestyle Post-surgery (HELP) intervention [181]. This project demonstrated the feasibility and high acceptability of accompanying this therapy using e-learning software suite for 10-week sections. Nevertheless, the participants of this project presented a significant WL (5.1%) achieved in the first three months post-intervention, justifying the high potential of this supportive therapy intervention for post-MS WR patients. However, studies with larger samples and better-designed methodologies are required to give this approach more weight in the future.

### 5.2. Pharmacotherapy

However, different studies have found that, besides lifestyle changes, PA levels, and behavioral therapies, the addition of pharmacotherapy to decrease appetite and elicit satiety is an effective alternative to interrupt WR [161,182]. In the last decade, great progress has been made with obesity pharmacotherapy with impressive results in weight loss. Interestingly, recent findings have found a particular link between weight-loss medication (WLM) and patients with RYGB and gastric banding, as increased WL in response to pharmacotherapy and the metabolic surgical procedures previously mentioned have been evidenced [169,172].

In this setting, a multicenter study with 319 patients demonstrated a WL ranging from ≥5% to ≥15% of their TBW in subjects with inadequate WR or WL after RYGB or LSG [183]. To evaluate the efficacy of the therapy, Hanipah et al. [169] set out to determine the effectiveness of phentermine, extended-release phentermine/topiramate, lorcaserin, and slow-release naltrexone/bupropion. After a year of adjuvant therapy, more than one-third of patients lost >5% of total weight. In addition, the study found that with higher BMIs, the efficacy between BMI at baseline and total weight loss at one year increased (*p* = 0.025).

Similarly, topiramate could be a cost-effective adjuvant in those patients with adjustable gastric banding (AGB) and difficulty in WL [184]. Recent studies investigated the efficacy of this drug and found a statistically significant association with WL, as patients receiving topiramate lost up to 10% of TBW [183]. In addition, topiramate + phentermine has also been tested, as it has been postulated in studies to be one of the most effective WLMs among patients [185].

In this context, Istfan et al. [186] studied the impact of topiramate + phentermine on WR after RYGB, finding that both their individual use and combination could significantly reduce WR after RYGB. Similar findings were found in studies that attempted to evaluate the efficacy of such a pharmacological combination [185,187]. Moreover, there is evidence that phentermine and fenfluramine compounds could be used as adjunctive therapy to inhibit appetite in bariatric surgical patients after 12 weeks of treatment [188].

In the same way, another adjuvant and anorexigenic drug indicated to treat low WL or WR post-MS is liraglutide [189,190]. This long-acting GLP-1 analogue induces WL by slowing gastric emptying and augmenting insulin secretion and satiety control via stimulating POMC/AgRP neurons in the hypothalamus [191,192]. In this respect, Wharton et al. [170] observed a significant WL (±7.7 kg, *p* < 0.05) in patients with post-RYGB WR, banding, and gastric sleeve who took 3.0 mg of liraglutide for 7.6 ± 7.1 months. These findings seem to be in line with other similar results from studies that sought to determine the efficacy of liraglutide as a short-term alternative treatment [193] and at high doses (3.0 mg) in post-surgical WR patients [194].

Multiple studies have been conducted using liraglutide as well as some with semaglutide [189,195]. After undergoing bariatric surgery, participants in another study began semaglutide treatment around a year later. Despite regaining 12% of their body weight since the surgery, the participants saw a remarkable 6% reduction in weight after only three months of semaglutide treatment [195].

Meanwhile, metformin has also been added to the list of possible adjuvant medications for post-MS WR, following the results of an intervention in patients with post-MS WR, who presented improvements in body weight and metabolic profile. In this regard, Torbay et al. [171] demonstrated that after the administration of 1800 ± 350 mg/day of metformin added to an ad libitum, low-carbohydrate, non-ketogenic, or high-protein diet (LCNK), patients with WR post-MS were able to reduce the equivalent of 5% of their weight, as well as reduce fasting blood glucose, HBA1c, cholesterol, and HOMA-IR levels, improving the insulin resistance that the patients had at the beginning of the study.

However, it is important to stress that better research involving larger samples with longer follow-ups and better evidence quality is still needed. Thus, determining the efficacy of adjuvant pharmacotherapy after metabolic surgery could be scaled to a more optimal and safe level, adding anti-obesity drugs for those patients who require such an alternative to control WR post-MS [182].

### 5.3. Endoscopic Interventions

On the other hand, endoscopic interventions represent a different approach to post-MS WR [163,196]. In this regard, a systematic review and meta-analysis evaluated the safety and efficacy of transoral outlet reduction (TORe) to treat post-RYGB WR on 850 patients. Moreover, WL ranging from 6.14 (kg) to 10.15 (kg) over 3 to 12 months was evidenced [196]. Furthermore, these results were consistent with other studies evaluating the efficacy of TORe using endoluminal suturing [164,197,198,199], positioning it as a feasible and promising endoscopic approach for patients with post-RYGB WR as well as for other complications such as SD [200].

Suturing techniques to improve TORe have also been studied, including full-thickness suture plus argon plasma mucosal coagulation (APC-TORe) and argon plasma mucosal coagulation alone (APMC-TORe) [201]. Although APMC-TORe usually demands several endoscopic sessions, the study indicated that both suturing techniques provide significant WL for patients with WR post-MS [201]. Similar findings were found in other studies evaluating the effectiveness of APC-TORe and APMC-TORe [173,174,175,202,203]. In this concern, modified endoscopic submucosal dissection (ESD) for TORe (ESD-TORe) has also been closely studied [204]. Interestingly, the combination of ESD-TORe and suturing provided a higher WL for patients with post-BGYR WR, with the ESD-TORe group experiencing a WL of 12.1% ± 9.3% versus 7.5% ± 3.3% for the traditional APC-TORe group.

### 5.4. Revisional Surgery

As a final point, although there are not enough studies on the subject, post-MS WR in some cases, may ultimately require revision or conversion surgeries, which are becoming more frequent every year [176,205,206]. Recently, Kermansaravi et al. [207] conducted a systematic review and meta-analysis that found that most revision surgical methods following primary RYGB have been safe and effective for post-MS WR. However, it is important to note that either distal Roux-enY gastric bypass (DRGB), biliopancreatic diversion with duodenal switch (BPD-DS), or single anastomosis duodenum-ileal bypass with gastric sleeve (SADI-S) seem to be the most effective long-term alternatives.

On the other hand, a prospective study demonstrated that revision procedures offered higher WLs after failed primary sleeve gastrectomy (SG) [176]. In this regard, investigators found an initial BMI of 41.9 (11.7) kg/m^2^ and subsequently, at follow-up, 12 months later, resulted in RYGB 13.2 (11.3) kg, and re-sleeve 12.0 (11.9) kg; *p* = 0.023. This 6.3 (5.1) kg/m^2^ decrease in BMI at one year provides evidence on the one hand of higher WL with revision bariatric surgery and, on the other hand, the preference of using techniques such as bypass to re-sleeve surgeries according to studies [176,177]. Likewise, other studies emphasize that the placement of silicone rings again induces greater WL in patients than in patients who have had the band since the original operation [179].

Although the data suggest optimal results, more studies with longer follow-ups are needed, especially those regarding post-MS revision surgery since it has been shown that the longer the post-operative follow-up, the better the results regarding post-MS WL [179]. Therefore, future studies should involve better methodology to achieve better evidence from less invasive approaches, such as lifestyle changes and behavioral therapies, to more invasive procedures, such as endoscopic interventions and revision surgery [60,178,208].

## 6. Conclusions

Both obesity and overweight are worldwide public health problems. The long-term results of traditional weight loss therapies, e.g., diet, exercise, and medication, especially those above, have not yet been thoroughly explored. On the other hand, MS is an effective treatment that improves and remits many related comorbidities. In addition, it favors sustained weight loss and improved patient quality of life while reducing mortality. Currently, the procedures included are Adjustable Gastric Banding, Vertical Banded Gastroplasty, Jejunoileal Bypass, Jejuno-colic Bypass, Roux-enY Gastric Bypass and Biliopancreatic Diversion with Duodenal Switch.

In conclusion, although MS is associated with many benefits, it also has multiple complications. One of the most prevalent is post-MS weight regain [209], caused by several psychological, behavioral, genetic, and anatomical factors. In addition, studies have also identified neuroendocrine, metabolic, and adaptive mechanisms that could influence WR in post-MS patients. Due to its multifactorial component [209], various therapies have been proposed to ameliorate WR. These include specialized multidisciplinary care (i.e., a combination of behavioral therapies, lifestyle changes, and weight control), appetite suppressants or anorexigenic pharmacotherapy, endoscopic procedures, and even revision surgery. However, it is worth noting that this is a subject with limited scientific evidence; therefore, further research is needed to elucidate the exact mechanisms involved in WR to establish an approach to combat the development of this complication of DS. These types of approaches are necessary and should be susceptible to periodic study.

## Figures and Tables

**Figure 1 jcm-13-01143-f001:**
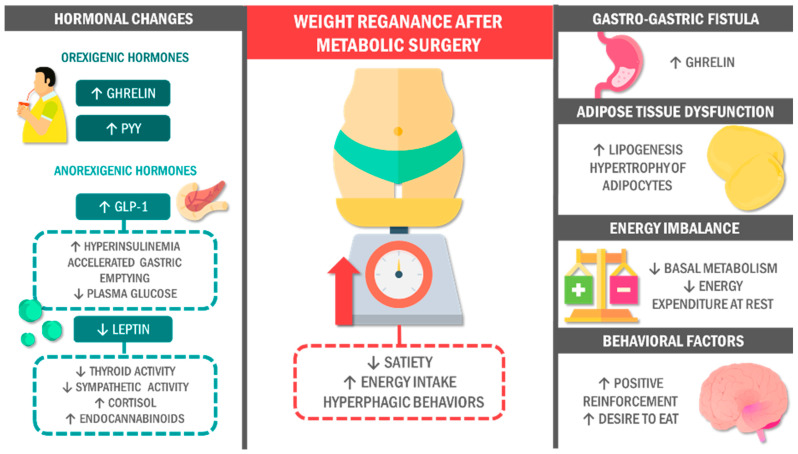
Potential mechanisms associated with weight regain after metabolic surgery. However, the surgery results are not always as expected. On the contrary, patients exhibit altered levels of orexigenic hormones that promote appetite and an excess of certain anorectic hormones that could lead to metabolic alterations associated with weight regain. Likewise, post-surgical dietary changes can be associated with developing an energy gap that favors excessive food consumption. In addition, post-MS patients tend to have maladaptive eating patterns capable of eliciting ingesting amounts of food similar to those before surgery. Abbreviations: GLP-1: glucagon-like peptide 1; PYY: peptide YY; MS: metabolic surgery.

**Table 1 jcm-13-01143-t001:** Lifestyle changes and pharmacotherapy of weight regain after bariatric surgery.

Authors (REF)	Strategies	Methodology	Results
Tettero et al. [159]	Lifestyle changes	Prospective CT-assessed changes in PA, WL and cardiorespiratory fitness	Improvements were seen in sport and leisure activity assessments (*n* = 3548, *p* < 0.001)
Gallé et al. [167]	Lifestyle changes	Non-randomized, controlled, prospective CT assessing a group of patients who chose to attend a 12-month lifestyle program	All the behavioral and physical outcomes improved significantly in the participants to the intervention.
Gils Contreras et al. [158]	Lifestyle changes	Prospective study assessed the extent to which increasing or decreasing adherence to the MedDiet assesed by Mediterranean Diet Adherence Screener	Individuals who increased adherence to MedDiet showed a significantly higher mean of total WL percentage 37.6% (35.5–39.8).
Stewart et al. [168]	Lifestyle changes	A meta-analysis assessed the effectiveness of behavioral interventions before and/or after BS in improving WL	Behavioral interventions in addition to MS appears to result in improved post-operative WL outcomes for people with severe obesity.
Hanipah et al. [169]	Phentermine, phentermine/topiramate extended release, lorcaserin, naltrexone slow-release/bupropion slow-release	Weight changes of 209 patients who received WL medications after MS were assessed in this CT.	37% lost > 5% of their total weight 1 year after pharmacotherapy was prescribed.
Wharton et al. [170]	Liraglutide	A CT assessed the effectiveness of liraglutide 3.0 mg in postbariatric surgery patients	Patients lost a significant amount of weight (−6.3 ± 7.7 kg, *p* < 0.05) regardless of the type of surgery they had (*p* > 0.05)
Torbay et al. [171]	Metformin	A CT assessed the impact of metformin on the weight of patients with WR after MS	Significant WL was observed in these patients (104.2 ± 2.4 vs. 99.4 ± 2.3 kg, *p* < 0.001)
Toth et al. [172]	Topiramate, phentermine, and metformin	Retrospective cohort study assessed the utility of common WL medications	WL medications are beneficial for WL in patients who have undergone MS

PA: physical activity; MedDiet: Mediterranean diet; MS: metabolic surgery; WL: weight loss; CT: clinical trial; WR: weight regain.

**Table 2 jcm-13-01143-t002:** Invasive strategies for prevention and management of weight regain after metabolic surgery.

Authors (REF)	Strategies	Methodology	Results
Jirapinyo et al. [164]	Endoscopic interventions	The retrospective study assessed the efficacy and predictors of long-term WL after TORe.	At 5 years after TORe, nearly all patients have cessation of WR, with the majority experiencing clinically significant WL.
Brunaldi et al. [173]	Endoscopic interventions	An RCT comparing the effectiveness of APC vs. FTS-APC for transoral outlet reduction	At 12 months, the mean %TWL was 8.3% ± 5.5% in the APC alone group versus 7.5% ± 7.7% in the FTS-APC group (*p* = 0.71).
de Quadros et al. [174]	Endoscopic interventions	An RCT compared APC to multidisciplinary management after WR.	Significant improvement in greater WL were found in the APC group. Both groups had similar WL (13.02 kg in the APC and 11.52 kg in the control).
Moon et al. [175]	Endoscopic interventions	A retrospective study was performed for 558 patients with WR after RYGB	Patients showed 6–10% total WL at 12 months.
Andalib et al. [176]	Revision surgeries	A single-centre retrospective study of patients who underwent revisional bariatric surgery for a failed previous SG.	Revisional procedures offer further WL after a failed primary SG. Bypass-type revisions are preferred over re-sleeve surgery.
Shimon et al. [177]	Revision surgeries	Patients who underwent post-LGS conversion to RYGB or DS were retrospectively analyzed.	At the last follow-up (>2 years), 15 RYGB patients had a reduction in BMI of 8.5–31.9 kg/m^2^ and 18 DS patients had a reduction in BMI of 12.8–31.9 kg/m^2^.
Jabbour et al. [178]	Revision surgeries	Systematic review which reported revision procedures on 1403 patients.	All methods of revision procedures have been effective. In the 1-year follow-up, DRGB presented a greater decrease in BMI.
Ferraz et al. [179]	Revision surgeries	Retrospective study assessed 29 patients who presented WR on follow-up after more than 5 years.	WL after revision surgery was greater in patients with longer revisional post-operative follow-up. Patients who underwent the placing of a silicon ring presented greater WL

RCT: randomized clinical trial; BMI: body-mass-index; SG: sleeve gastrectomy; TORe: Endoscopic Transoral Outlet Reduction; RYGB: Roux-en-Y gastric bypass; DRGB: distal Roux-en-Y gastric bypass; APC: Argon plasma coagulation; FTS-APC: endoscopic full-thickness suturing; DS: duodenal switch; WL: weight loss; WR: weight regain.

## Data Availability

No new data were created or analyzed in this study. Data sharing is not applicable to this article.

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
