# Peer review of "Weight Regain after Metabolic Surgery: Beyond the Surgical Failure"

_jcm, 2024, doi:10.3390/jcm13041143_

Round 1

Reviewer 1 Report

Comments and Suggestions for Authors

FIGURE 1 IS REPEATED, I SUGGEST THAT THE ONE THAT APPEARS SECOND IN THE WRITING REMAINS.

FIGURE 2 APPEARS CROPPED, I SUGGEST EITHER REDUCING ITS SIZE OR CHANGING IT FROM VERTICAL TO HORIZONTAL.

OTHERWISE THE INSTRUMENT IS WELL DESCRIBED AND THE CONCLUSIONS ARE VALUABLE AND USEFUL TO MANY IN THE FIELD OF INTEREST OF THIS REVIEW ARTICLE.

THESE TYPES OF APPROACHES ARE NECESSARY AND SHOULD BE SUSCEPTIBLE TO PERIODIC STUDY, BOTH TO SOCIALIZE (SPREAD) AND ALWAYS TEND TO SEEK THE QUALITY OF LIFE OF THIS TYPES OF PATIENTS.

Author Response

FIGURE 1 IS REPEATED, I SUGGEST THAT THE ONE THAT APPEARS SECOND IN THE WRITING REMAINS.

A: We've unified Figure 1. We sorry for the mistake

FIGURE 2 APPEARS CROPPED, I SUGGEST EITHER REDUCING ITS SIZE OR CHANGING IT FROM VERTICAL TO HORIZONTAL.

A: We have sent the figure in a modifiable format for editing by the editorial team as they consider

OTHERWISE THE INSTRUMENT IS WELL DESCRIBED AND THE CONCLUSIONS ARE VALUABLE AND USEFUL TO MANY IN THE FIELD OF INTEREST OF THIS REVIEW ARTICLE.

A: Thanks for your comment

THESE TYPES OF APPROACHES ARE NECESSARY AND SHOULD BE SUSCEPTIBLE TO PERIODIC STUDY, BOTH TO SOCIALIZE (SPREAD) AND ALWAYS TEND TO SEEK THE QUALITY OF LIFE OF THIS TYPES OF PATIENTS.

A: This sentence was added to conclusion

Reviewer 2 Report

Comments and Suggestions for Authors

The present review deals with weight regain after  metabolic surgery.

 Comments:

1.      Although there are plenty of previous papers dealing with weight regain and bariatric surgery, the present review is  interesting and useful to clinicians

2.      Introduction is not focused on the purpose of the review.

3.      Table 1 is not relevant to the topic and can be removed. Complications (other than weight regain) could be mentioned briefly in the text.

4.      Table 2 and Table 3 could be shortened; results of the studies could be presented in a more coded way.

5.      There are plenty of previous articles within 2023 dealing with weight regain and bariatric surgery. References could be updated.

Comments on the Quality of English Language

Minor editing of English language is required.

Author Response

  1. Although there are plenty of previous papers dealing with weight regain and bariatric surgery, the present review is  interesting and useful to clinicians

A: Thanks for your comment

  1. Introduction is not focused on the purpose of the review.

A: We edited the introduction to focused the purpose

  1. Table 1 is not relevant to the topic and can be removed. Complications (other than weight regain) could be mentioned briefly in the text.

A: The table 1 was deleted

  1. Table 2 and Table 3 could be shortened; results of the studies could be presented in a more coded way.

A: We've slightly reduced the content of the tables. However, we consider that they should be explanatory since the characteristics of patients in each study are specific

  1. There are plenty of previous articles within 2023 dealing with weight regain and bariatric surgery. References could be updated.

A: We add the references: Front. Endocrinol. 2023;14:1267014. doi: 10.3389/fendo.2023.1267014     Curr Diab Rep. 2023; 23(3): 31–42.           https://doi.org/10.1016/j.obmed.2023.100528

Reviewer 3 Report

Comments and Suggestions for Authors

The article highlights a very important problem of weight regain after metabolic surgery. It is the overview of the causes of weight regain and the possibilities of their correction. The manuscript is very interesting and relevant for the field and presented in a well-structured manner. The paper is of interest to a wide range of specialists who deal with the problem of obesity.

There are minor inaccuracies in the design of the article. Figure 1 (Mechanisms associated with weight regain after metabolic surgery) is duplicated on pages 11 and 12. The only difference between the figures is the phrase “delayed gastric emptying” on page 11 and “accelerated gastric emptying” on page 12 as an effect of GLP-1 increase. The tables 2 and 3 have equal titles, which makes it difficult to comprehend the text.

Overall, the article is excellent.

Author Response

The article highlights a very important problem of weight regain after metabolic surgery. It is the overview of the causes of weight regain and the possibilities of their correction. The manuscript is very interesting and relevant for the field and presented in a well-structured manner. The paper is of interest to a wide range of specialists who deal with the problem of obesity.

A: Thanks for your comment

There are minor inaccuracies in the design of the article. Figure 1 (Mechanisms associated with weight regain after metabolic surgery) is duplicated on pages 11 and 12. The only difference between the figures is the phrase “delayed gastric emptying” on page 11 and “accelerated gastric emptying” on page 12 as an effect of GLP-1 increase. The tables 2 and 3 have equal titles, which makes it difficult to comprehend the text.

A: We've unified Figure 1. We sorry for the mistake. Table 2 and 3 titles were edited

Overall, the article is excellent.

Round 2

Reviewer 2 Report

Comments and Suggestions for Authors

The revised version of the manuscript has been improved.